# End-of-life care in hematological malignancies – a nationwide comparative study on the Swedish Register of Palliative Care

Ellen Skåreby[1], Per Fürst[2,3], Lena von Bahr[4,5]*

**1** Department of Internal Medicine, Kungälv Hospital, Sweden, **2** Research Department Palliative Care, Stockholm Sjukhem Foundation, Stockholm, Sweden, **3** Department of Neurobiology, Care Sciences and Society, Karolinska Institute, Stockholm, Sweden, **4** Department of Medicine, Sahlgrenska Academy, Gothenburg University, Sweden, **5** Section of Hematology and Coagulation, Sahlgrenska University Hospital, Gothenburg, Sweden

* lena.von.bahr@gu.se

## Abstract

### Background

Patients with hematological malignancies are less likely to be referred to specialized palliative care, and more likely to receive aggressive end-of-life care than patient with solid tumors. The Swedish Register of Palliative Care (SRPC) collects end-of-life care quality data from a majority of health facilities in Sweden. We here use the national data from the SRPC to evaluate the quality of end-of-life care in patients with hematological malignancies in Sweden.

### Methods

In a retrospective, observational registry study all adult registered cancer deaths in the years 2011–2019 were included. For the main analysis, patients with unexpected deaths or co-morbidities were excluded. Descriptive statistics and multiple logistic regression, adjusting for age and sex, were used.

### Results

A total of 119 927 patients were included, 8 550 with hematological malignancy (HM) and 111 377 with solid tumor (ST), corresponding to 43% of all deaths due to HM and 61% of ST deaths during the observed period.

Significantly more ST patients than HM received end-of-life care in a specialized palliative unit (hospice, palliative ward or specialized home care), 54% vs 42% (p<0.001), and this difference could be seen also in the very old (80+). End-of-life care quality measures were significantly worse for HM patients than ST patients, which could partly be explained by the lower receipt of specialized palliative care. The most common symptom in both groups were pain, followed by anxiety. HM patients were less

**Data availability statement:** The data contain potentially identifying information regarding individuals and therefore are subject to ethical and legal restriction to public sharing. We cannot share the data set as it is not permitted by Swedish law and the ethical permission obtained only allows for public sharing on a group level. Data are available from the Swedish Register of Palliative Care (info@palliativregis-tret.se) for researchers after appropriate ethical review.

**Funding:** This study received funding from Blodcancerfonden : Blodcancerfonden as a funding agent had no role in study design, data collection and analysis, decision to publish, or preparation of the manuscript. Patient representatives mediated by Blodcancerfonden were involved in study design.

**Competing interests:** The authors have declared that no competing interests exist.

likely to achieve complete symptom relief (p<0.001) which appears to be related to the receipt of specialized palliative care.

## Conclusion

Patients with hematological malignancies are more likely to die in emergency hospital and less likely to receive specialized palliative competence in end-of-life. This also translates into less qualitative end-of-life care and less efficient symptom relief.

## Introduction

Despite several advances in the treatment of hematological malignancies in recent years, the long-term survival is still below 50% for most hematological diagnoses [1,2]. Both disease and treatments carry a significant symptom burden [3] and when referred to palliative care their palliative needs are similar to those of patients with solid tumors [4–7]. However, patients with hematological malignancies are less likely to be referred to palliative care [8,9], are more likely to receive aggressive treatment at end of life [10,11] and to die in an emergency hospital [9,11] than patients with solid malignancies.

Hematological malignancies are complex and heterogenous. They can differ from solid tumors with more unpredictable disease trajectories [12,13] and a possibility of cure or long-term survival can remain even in advanced or relapsed disease. Traditional quality measures of end of life care can therefore be difficult to apply to patients with hematological malignancies, as it might be concordant with good practice and patients' wishes to pursue aggressive treatment until end of life [14,15]. This perspective gets lost when studies focus on health care utilization at end of life, and studies evaluating the actual benefits of palliative care interventions do not address the end of life care provided in hospitals [16].

The Swedish Register of Palliative Care (SRPC) is a unique source of national data regarding the end of life care provided regardless of health care setting or diagnosis [17], with a coverage of close to 80% of patients dying of cancer in Sweden. In this study, we use data from the SRPC to map end of life care quality specifically for patients with expected deaths, comparing hematological malignancies with solid tumors.

## Methods

The Strengthening the Reporting of Observational Studies in Epidemiology (STROBE) criteria has been used to report the methods and result.

### Design

This was a retrospective observational registry study based on the Swedish Register of Palliative Care (SRPC). The SRPC contains answers to the End-of-life questionnaire (S1 Appendix) which is recorded by health care providers after a patients' death and focuses on quality of care and symptoms during the last 7 days of life. SRPC also contains registered primary cause of death according to the 10th revision of the International Classification of Diseases (ICD10), retrieved from the Cause of Death Register of the Swedish National Board of Health and Welfare. Data was

retrieved in pseudonymized form from the SRPC on the 23rd of May 2023, and individual patients could not be identified by the researchers.

The study plan was pre-registered on Open Science Framework, https://doi.org/10.17605/OSF.IO/VMTSR

The study was approved by the Swedish Ethical Review Authority (Dnr 2023-03378-01). As all individuals registered were deceased, they could not consent to participating in the study and the need to obtain consent was waived by the Ethical Review Authority.

## Participants

Data was collected for all registered deaths with patient age > 18 years for the nine consecutive years of 2011–2019.

Cases with ICD codes C00-C96 + D45-D47 were extracted for analysis and divided into two study cohorts: hematological malignancy (HM) (ICD C81-C96 + ICD D45-D47) and solid tumors (ST) (ICD C00-C80).

## Exclusion criteria

Patients with unexpected death or co-morbidities contributing to death were excluded from the main analyses. Contributing co-morbidities are recorded in the SRPC in the following categories: Cardiac disease, Respiratory disease, Cognitive disorder (dementia), Stroke, Other neurological disease, Diabetes, Fracture, Infection, Multi-morbidities, Other co-morbidities. Separate analyses of these cases were made to evaluate validity of the results and are presented in the results section.

## Variables

Outcome variables were place of death and quality indicators of end-of-life care, as documented in the SPRC. These included conversations about transition to end-of-life care, administration of intravenous fluids during the last 24 hours, symptom relief etc. If not otherwise specified, the variables recorded in the SRPC are recorded for the last 7 days of life. A complete summary of included variables can be found in S1 Table.

Age and sex were included as potential confounding variables. Subgroup analyses were performed based on place of death, hematological diagnosis, and age strata.

## Statistics and sample size estimation

All outcome variables were analyzed using multiple logistical regression (applying the R glm() function with family=binomial) adjusted for age and sex if not otherwise specified. Odds ratios were derived from the logistical regression and presented with 95% confidence intervals. Baseline characteristics were compared with students t-test or ANOVA and chi2 test for continuous and categorical data respectively.

All analyses were made using R software [18] version 4.4.0, with R Studio, R Markdown and packages, forestploter and ggplot2 for tables and figures.

Sample size was determined by the available register data, aiming for a large population with representative data. Starting year 2011 was determined by the availability of data for cause of death. Ending year 2019 was determined to avoid influence of the covid-19 pandemic.

## Missing data

There were no missing data related to the grouping variables (ICD code, Place of death, Age) or confounder variables (Age, Sex).

There were varying degrees of missing values in the outcome variables, where the questionnaire had the option "Don't know". A multivariate correlation analysis showed significant correlation between diagnosis cohort and frequency of missing values for several missing outcome variables, as well as a high correlation between place of death and all missing outcome values. As the missing values can be considered Missing Not At Random (MNAR) [19] we decided to forgo any imputations and present the results as a complete case analysis.

The frequencies of missing values are presented for each variable in S1 Table.

## Results

A total of 14 673 patients with HM and 145 555 with ST as cause of death were identified in the SPRC for the defined period, which represents 74% and 79% of all deaths recorded by the Swedish National Board of Health and Welfare for the diagnoses respectively (p < 0.001). After excluding cases with unexpected death and with co-morbidities contributing to death, the resulting study cohorts consisted of 8 550 patients with HM and 111 377 with ST (Fig 1).

Demographical variables are presented in Table 1.

### Place of death and access to specialized palliative care

The most common place of death in HM was an emergency hospital (38%), followed by hospice or palliative ward (27%) and own home (21%). For ST, the most common place of death was hospice or palliative ward (36%), followed by own home (26%) and emergency hospital (21%) (Table 2).

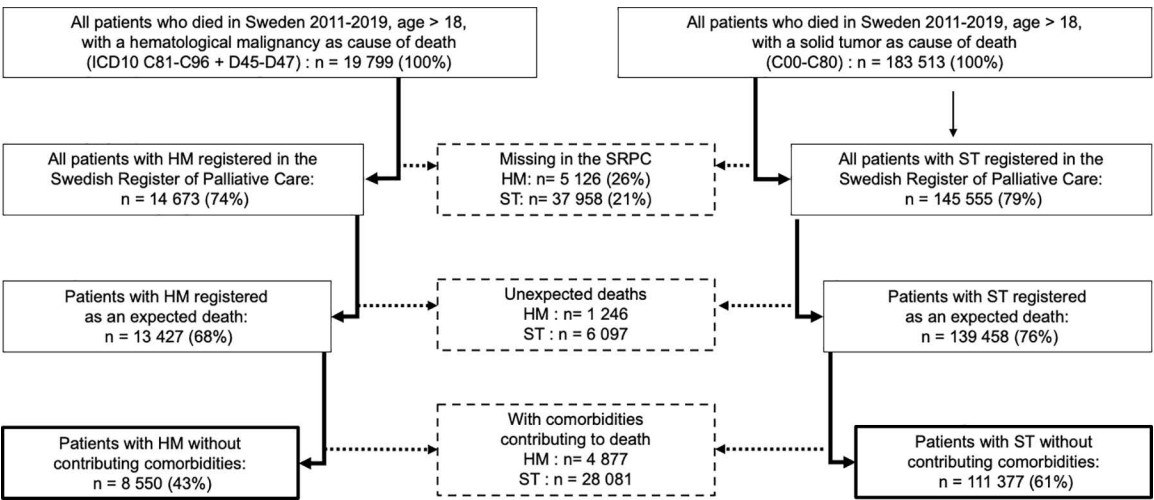

**Fig 1. Flowchart of case selection** Percentages refer to all deaths registered by the Swedish National Board for Health and Welfare, age >18, for the specified cause of death.

**Table 1. Demographical variables.**

|  | HM (N=8 550) | ST (N=111 377) |
|---|---|---|
| Age | p < 0.001 | |
| Median (range) | 77 [18,103] | 74 [18,104] |
| Sex | p < 0.001 | |
| Female | 45% | 50% |
| Male | 55% | 50% |
| Age intervals | p < 0.001 | |
| <30 | 58 (1%) | 352 (0%) |
| 30-40 | 68 (1%) | 949 (1%) |
| 40-50 | 172 (2%) | 3209 (3%) |
| 50-60 | 454 (5%) | 9369 (8%) |
| 60-70 | 1467 (17%) | 24874 (22%) |
| 70-80 | 2953 (35%) | 36915 (33%) |
| >80 | 3378 (40%) | 35709 (32%) |

**Table 2. Place of death and access to specialized palliative care.**

| | All patients | | Patients aged >80 | |
|---|---|---|---|---|
| | HM (N=8 550) | ST (N=111 377) | HM (N=3 012) | ST (N=32 253) |
| Place of death | p < 0.001 | | p < 0.001 | |
| Own home | 21% | 26% | 20% | 23% |
| Nursing home | 14% | 17% | 23% | 30% |
| Emergency hospital | 38% | 21% | 30% | 17% |
| Hospice/ palliative in-patient care | 27% | 36% | 27% | 29% |
| Other | 1% | 1% | 0% | 1% |
| Access to specialized palliative care | p < 0.001 | | p < 0.001 | |
| In specialized palliative unit* | 42% | 54% | 39% | 43% |
| Other#, with palliative team consulted | 9% | 14% | 8% | 15% |
| None | 48% | 31% | 52% | 43% |

*Hospice/ palliative in-patient care OR Own home with support of specialized palliative team

#Any place of death apart from specialized palliative unit

Access to specialized palliative care (SPC) was significantly lower for HM than for ST patients, both for dying in a specialized palliative unit and for palliative team consults in other places of death. The risk of dying in an emergency hospital was significantly higher in HM. These differences could be seen also in the very old (80+) population (Fig 2).

There is a trend over the studied period for increased access to SPC as well as for a decrease in deaths in emergency hospital, both in the HM and the ST groups (Fig 3a and 3b).

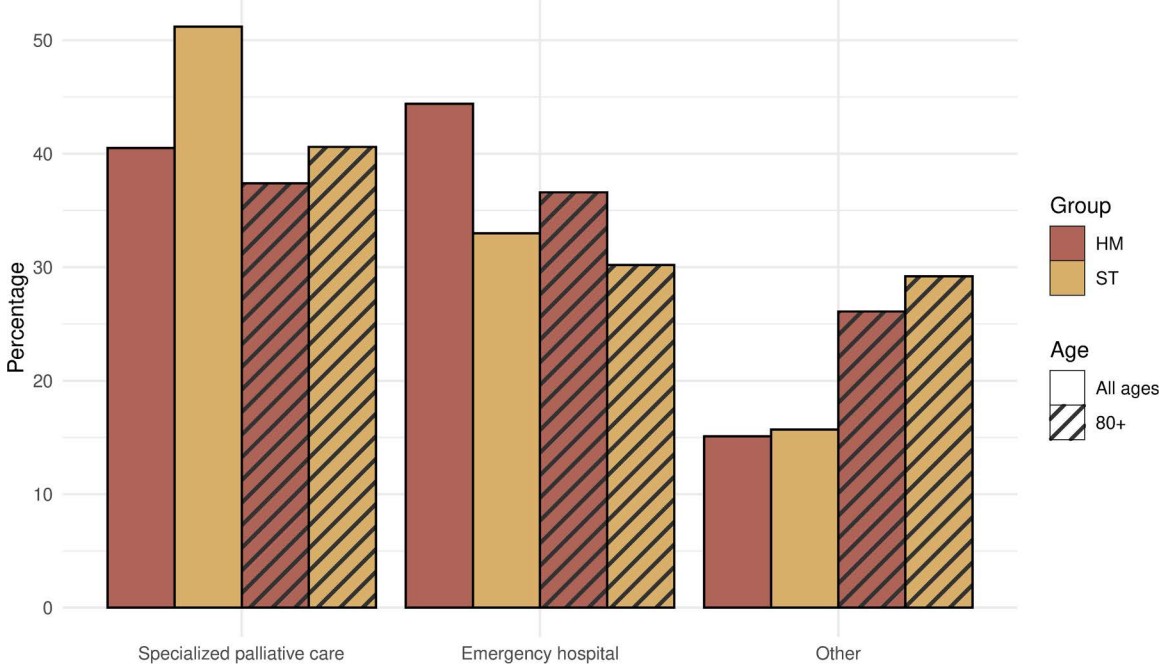

**Fig 2. Deaths in SPC unit or emergency hospital** Specialized palliative care = Hospice/ palliative in-patient care OR Own home with support of specialized palliative team.

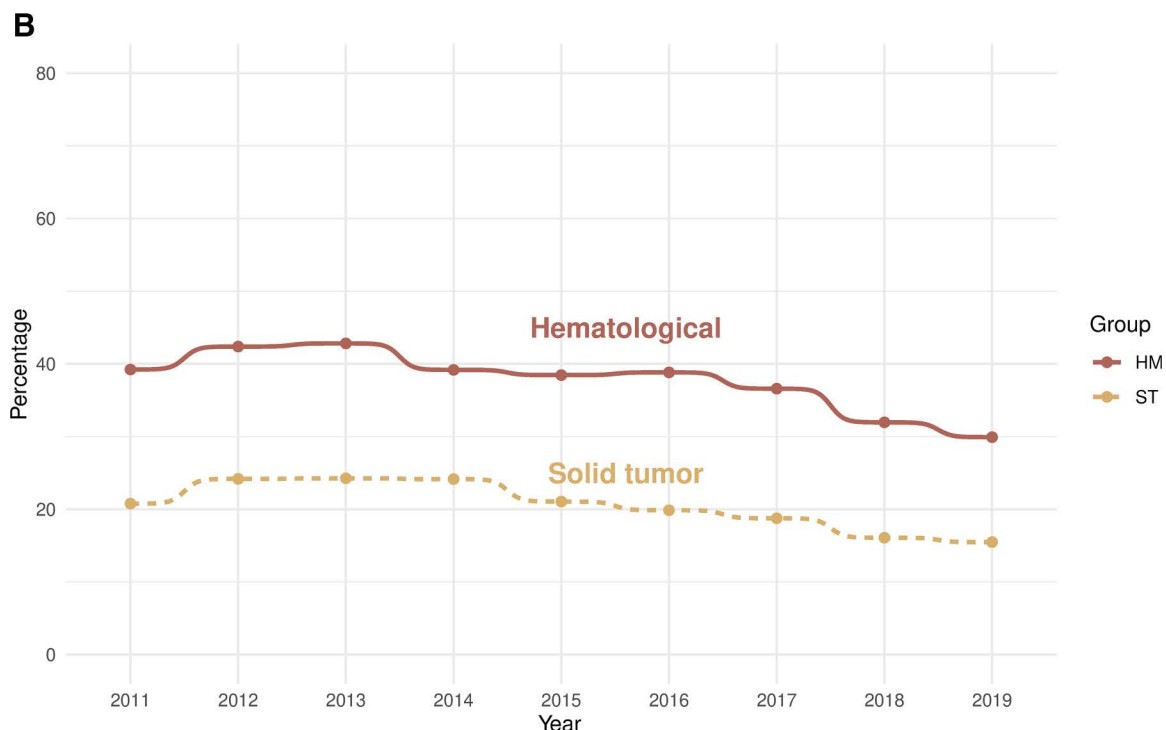

**Fig 3. Trends over time; 3a**: SPC access, **3b**: Deaths in emergency hospital.

The main hematological diagnoses in this study were lymphoma, myeloma, and acute myelocytic leukemia (AML). The distribution of place of death and access to palliative care varied somewhat between the groups, as can be seen in Table 3 (S2 Table for all diagnoses including lymphoma subgroups). Emergency hospital was the most common place of death and access to SPC was lower than for the ST group for all the HM diagnoses.

## End-of-life care quality indicators

There were statistically significant differences between the HM and ST groups on all quality indicators analyzed, except for someone present at time of death, with a consistently worse result for the HM patients (Table 4). The differences could generally be seen also in the very old (80+).

The most pronounced differences were seen in knowing the preference for place of death, administration of iv fluids during the last 24 hours and the use of validated methods for evaluation of pain.

## Symptoms and symptom relief

The frequencies of break-through symptoms recorded in the last 7 days of life are shown in Table 5, together with the percentage achieving complete relief.

HM patients are recorded to experience a median of 2 symptoms (range 0–6), while ST patients have a median of 3 (range 0–6), with pain being the most frequent in both groups followed by anxiety. The ST patients have a higher OR for their symptoms being completely relieved with the exception of nausea and rattles where there was no difference.

Dyspnea as breakthrough symptom is also associated with iv fluids in the last 24 hrs (OR 1.95, p < 0.001) and when administration of iv fluids is included as a variable in the multiple analysis there is no significant difference between HR and ST.

For all hematological diagnoses, pain was the most common symptom ranging from 72% in CLL patients to 83% in the myeloma patients. The frequency of breakthrough symptoms by hematological diagnosis is shown in Fig 4.

## Impact of place of death and access to specialized palliative care

As death in an emergency hospital was substantially more common for patients with HM, and access to SPC conversely lower, we further analyzed what association these factors had with the palliative care quality indicators as well as with symptoms and symptom relief.

**Table 3. Place of death and SPC access by main hematological diagnoses.**

|  | Lymphoma (N=3 092) | Myeloma (N=1 956) | AML (N=1 804) | MDS (N=549) | CLL (N=539) |
|---|---|---|---|---|---|
| Age | p < 0.001 | | | | |
| Median (range) | 77 [18,101] | 75 [21,99] | 76 [19,98] | 78 [29,99] | 79 [21,99] |
| Place of death | p < 0.001 | | | | |
| Own home | 19% | 20% | 23% | 19% | 19% |
| Nursing home | 16% | 15% | 7% | 20% | 17% |
| Emergency hospital | 35% | 38% | 43% | 36% | 43% |
| Hospice/ palliative in-patient care | 29% | 27% | 26% | 25% | 21% |
| Other | 1% | 0% | 1% | 1% | 0% |
| Access to specialized palliative care | | | p < 0.001 | | |
| In specialized palliative unit* | 42% | 40% | 45% | 45% | 32% |
| Other#, with palliative team consulted | 10% | 10% | 9% | 7% | 7% |
| None | 47% | 49% | 47% | 48% | 60% |

*Hospice/ palliative in-patient care OR Own home with support of specialized palliative team

#Any place of death apart from specialized palliative unit

**Table 4. End-of-life care quality indicators.**

| | All patients | | | Patients aged >80 years | | |
|---|---|---|---|---|---|---|
| | HM (N=8 550) | ST (N=111 377) | OR* 95% CI | HM (N=3 012) | ST (N=32 253) | OR* 95% CI |
| Communication | | | | | | |
| Documented shift to end-of-life care | | | | | | |
| Yes | 90% | 91% | **0.68** 0.62, 0.75 | 89% | 91% | **0.64** 0.54,0.76 |
| Documented conversation with patient about transition to end-of-life care | | | | | | |
| Offered | 82% | 87% | **0.68** 0.64,0.72 | 80% | 85% | **0.70** 0.64,0.78 |
| No | 17% | 12% | | 18% | 14% | |
| Unable to participate | 2% | 1% | | 2% | 1% | |
| Documented conversation with next of kin about transition to end-of-life care | | | | | | |
| Offered | 91% | 92% | 0.91 0.84,1.00 | 90% | 91% | 0.90 0.78,1.03 |
| No | 8% | 7% | | 9% | 8% | |
| No known relations | 1% | 1% | | 1% | 1% | |
| Patients' preference for place of death was known | | | | | | |
| Yes | 65% | 74% | **0.66** 0.63,0.69 | 66% | 73% | **0.72** 0.66,0.79 |
| Someone present at time of death | | | | | | |
| Yes | 87% | 88% | 0.95 0.89,1.02 | 83% | 85% | **0.86** 0.78,0.96 |
| Next of kin offered a follow-up talk | | | | | | |
| Yes | 87% | 89% | **0.84** 0.78,0.90 | 84% | 86% | **0.87** 0.78,0.98 |
| MEDICAL | | | | | | |
| Parenteral fluids in last 24 hrs | | | | | | |
| Yes | 17% | 11% | **1.66** 1.56,1.76 | 11% | 7% | **1.63** 1.44,1.84 |
| Pain evaluated with validated methods | | | | | | |
| Yes | 46% | 53% | **0.75** 0.72,0.79 | 45% | 49% | **0.84** 0.78,0.91 |
| Other symptoms evaluated with validated methods | | | | | | |
| Yes | 25% | 30% | **0.81** 0.77,0.86 | 26% | 27% | 0.93 0.85,1.01 |
| Prescription of PRN drugs | | | | | | |
| Opioids for pain | 97% | 98% | **0.66** 0.58,0.75 | 97% | 98% | **0.76** 0.62,0.95 |
| Drugs for anxiety | 94% | 95% | **0.73** 0.67,0.80 | 93% | 94% | **0.72** 0.65,0.80 |
| Drugs for nausea | 85% | 90% | **0.64** 0.60,0.68 | 84% | 88% | **0.83** 0.72,0.97 |
| Drugs for rattles | 91% | 94% | **0.72** 0.66,0.78 | 92% | 93% | **0.86** 0.75,0.99 |

*Odds ratios with correction for age and sex. Bold indicates statistical significance.

**Table 5. Symptoms and symptom relief.**

| | HM (N=8 550) | ST (N=111 377) | OR HM* 95% CI |
|---|---|---|---|
| Pain | | | |
| Frequency | 78% | 83% | **0.75** 0.72,0.80 |
| Completely relieved | 74% | 76% | **0.86** 0.81,0.91 |
| Dyspnea | | | |
| Frequency | 25% | 24% | **1.10** 1.04,1.16 |
| Completely relieved | 40% | 43% | **0.86** 0.78,0.94 |
| Anxiety | | | |
| Frequency | 58% | 59% | 1.00 0.95,1.05 |
| Completely relieved | 64% | 68% | **0.81** 0.76,0.87 |
| Nausea | | | |
| Frequency | 18% | 24% | **0.69** 0.65,0.73 |
| Completely relieved | 58% | 58% | 0.98 0.88,1.10 |
| Confusion | | | |
| Frequency | 30% | 28% | **1.10** 1.05,1.16 |
| Completely relieved | 22% | 24% | **0.89** 0.80,0.98 |
| Rattles | | | |
| Frequency | 47% | 51% | **0.86** 0.82,0.90 |
| Completely relieved | 47% | 47% | 0.99 0.92,1.05 |

*Odds ratios with correction for age and sex. Bold indicates statistical significance.

When adding SPC access to the multivariate analysis the OR for HM as cause of death shifted towards 1 for all parameters where HM was shown to be significantly worse than ST before adjusting (Fig 5). In Fig 6, the OR for hospital as place of death and SPC access, respectively, can be seen for each of the outcomes.

### Unexpected deaths and co-morbidities

7 323 patients (1 246 with HM and 6 097 with ST, significantly more in the HM group, p < 0.001) were recorded as unexpected deaths and were excluded from the main analysis. The patients with unexpected deaths were more likely to have co-morbidities contributing to death (60% of unexpected vs 22% of expected, p < 0.001) and the majority (62%) died in an emergency hospital (p < 0.001). End-of-life care outcomes were not reported to the SRPC for unexpected deaths.

32 958 patients (4 877 with HM and 28 081 with ST, significantly more in the HM group, p < 0.001) of patients with expected deaths were recorded as having co-morbidities contributing to death and were excluded from the main analysis.

The most common co-morbidities were cardiac disease and multimorbidity, the frequency of all registered co-morbidities can be seen in S3 Table.

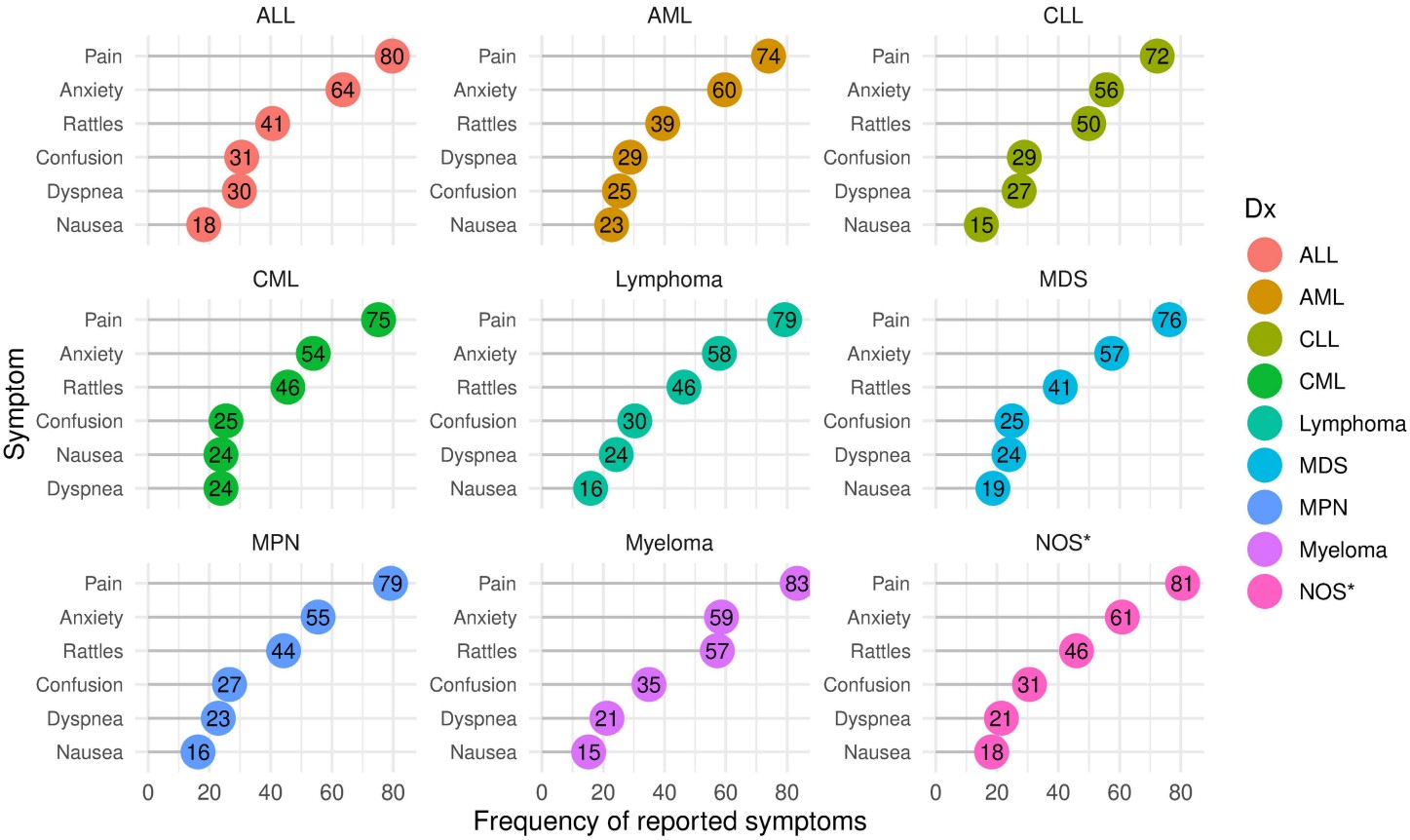

**Fig 4. Lollipop plot of symptoms by hematological diagnosis.** Frequency of symptoms reported during the last week of life, by hematological diagnosis. *NOS = hematological malignancy, not otherwise specified.

Patients with co-morbidities were significantly older (median age 82 vs 74 in those without co-morbidities, p < 0.001) and the most common place of death was a nursing home (40%). They had significantly lower access to SPC (29% vs 66% in those without co-morbidities, p < 0.001) and showed generally worse end-of-life care outcomes even after adjusting for SPC access.

## Discussion

In this study, we confirm previous findings of health care utilization at end of life for HM [9], with emergency hospital as the most common place of death. HM patients also showed worse quality for most recorded end of life care outcomes compared to ST. We restricted the analysis to cases where death was recorded as expected, and in the vast majority of cases there was a documented decision about transition to end of life care, which might suggest that the differences were not due to patients dying during treatment with curative intent. The same patterns could also be seen in the very old (80+), where HM are to be considered incurable in the majority of cases.

In addition, we show that break-through symptoms at end of life are common in HM, regardless of place of death. Relief of pain, dyspnea and anxiety was slightly but significantly less in HM patients compared to ST, a difference that was mainly explained by the lower access to palliative care. A worrying find was the higher administration of iv fluids in HM patients, known to increase risk of dyspnea [20,21], and could also in this material be shown to correspond to a significantly higher risk of dyspnea at end of life. It seems that the end-of-life care provided to HM patients, especially in the emergency hospital

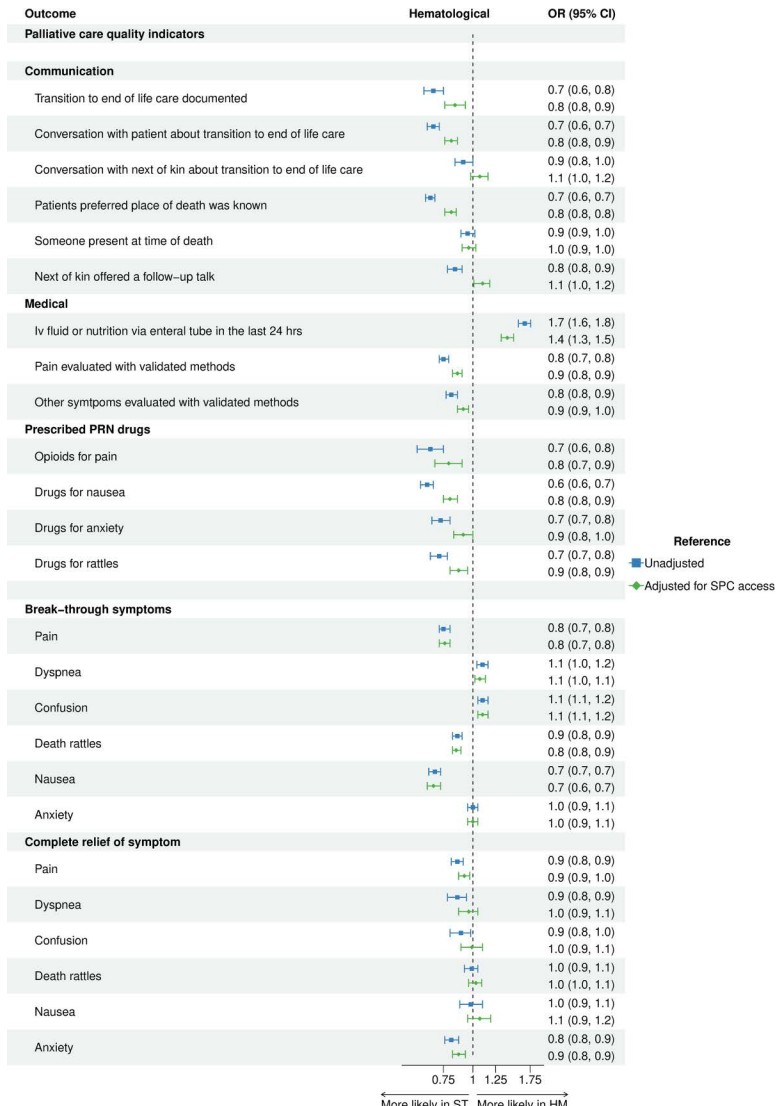

**Fig 5. Association of HM with palliative care quality indicators.** Forest plot with OR for HM as cause of death, with and without adjustment for SPC access.

setting, is falling short of providing the symptom relief that could have been possible in a specialized palliative care setting. This confirms earlier studies on end-of-life care quality in the hospital setting, without focus on HM [22,23].

A weakness in the study is that though all deaths were recorded as expected, there is no information regarding when the transition to end-of-life care took place. In HM, transitions to end-of-life care commonly happens late [14,24] which is one of several obstacles to specialized palliative care in this patient group [15]. However, most end-of-life care quality outcomes reported here focus on the very end of life (last 24 hours for administration of iv fluids) or should be in harmony with providing care to severely ill patients in a curative setting as well (evaluating pain and other symptoms with validated methods).

Another weakness is possible quality problems in the recorded register data. SRPC has validated the EOL questionnaire with high consistency in specialized palliative care [25] but the quality of recorded data was found to be lower in

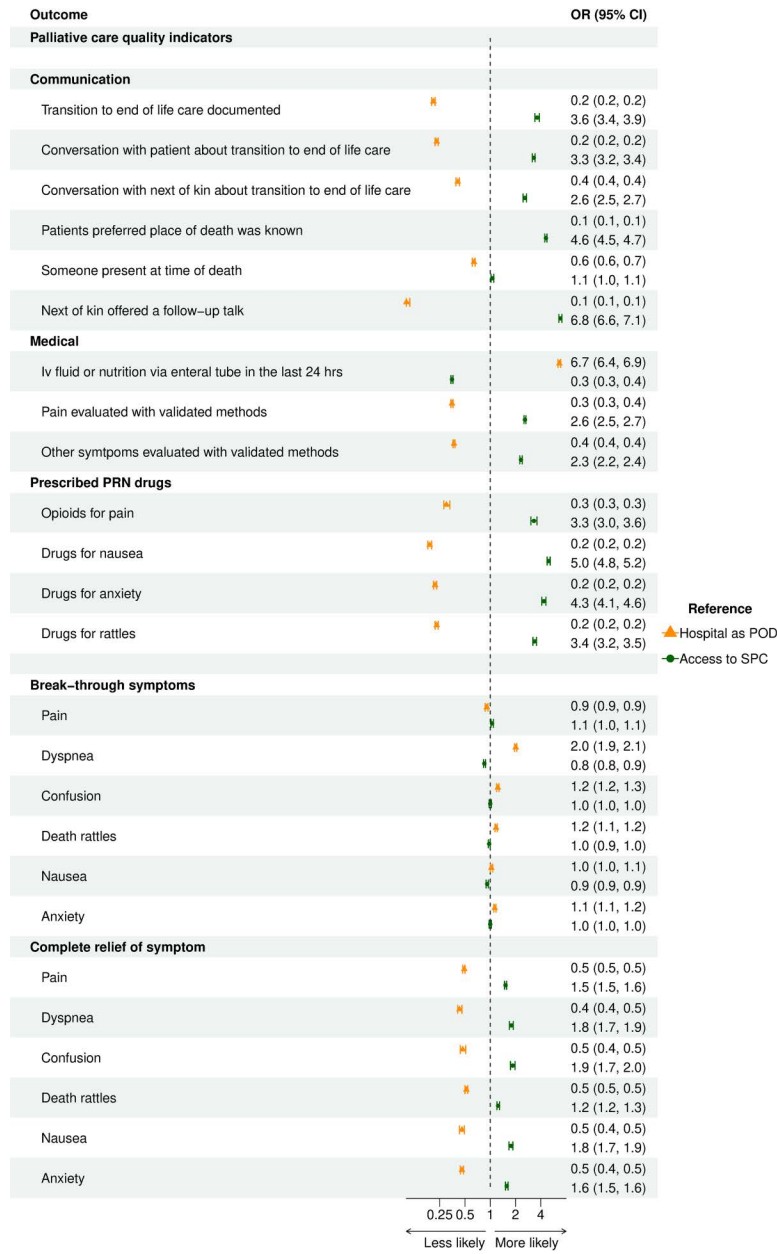

**Fig 6. Association of place of death with palliative care quality indicators.** Forest plot with OR for Hospital as place of death, and for SPC access, respectively.

emergency hospitals [26]. Missing data (with "don't know" answers) for end-of-life care outcomes in our data were also more common in emergency hospitals as well as in HM, and HM patients were to a lesser degree registered in the SRPC (Fig 1). It is however unlikely that these missing data would be responsible for the significant differences demonstrated in this study, as lower quality of documenting and reporting palliative care outcomes are more likely to be associated with lower quality of care than the opposite.

Despite inadequate end-of-life care in HM has been reported for many years [10] there are many obstacles to overcome in providing better access to SPC. Efforts are being made to facilitate earlier introduction of palliative care [27–30], to increase quality of life during aggressive treatment as well as to improve end-of-life care. Surveys and interview studies with hemato-oncology and palliative care practitioners repeatedly report that SPC being associated with end-of-life care is a hindrance to earlier introduction [14,31] and that SPC services are unable to provide some aspects of care that are essential for HM patients even in late stages of the disease, such as transfusions [15,24,32]. It is possible that emergency hospital will continue to be the main place of death in HM, and maybe rightfully so in view of the complex needs and the close bond and high confidence HM patients often express towards the hematology clinic [24]. But given the obvious quality differences revealed in this study, the competence in addressing palliative needs and providing high standard end-of-life care needs to be raised in the hematological departments. An earlier palliative approach, in parallel with active treatment and with an open, continuous discussion regarding plans and hopes in the eventuality of treatment failure could be a way to achieve this. WHO has deemed access to palliative care a human right [33] that should be possible to provide also in emergency hospital settings.

There is a somewhat hopeful trend of increased SPC access over the years captured in this study, deliberately cut at 2019 to avoid confusing data with the enormous impact of the covid-19 pandemic on all aspects of health care, palliative care not the least. In Sweden the many covid-related deaths in nursing homes led to a heated debate where palliative care was equated with "letting the old and frail die without treatment" [34]. It will be interesting to see if the positive trend in SPC access preceding the pandemic can be resumed in later years.

In conclusion, this study confirms previous reports of sub-optimal end-of-life care in HM patients, in a nationwide register study covering a majority of all cancer deaths in Sweden over a period of nine years. These differences cannot entirely be explained by the complex and sometimes unexpected disease trajectories of HM and emphasizes the need to improve competence in palliative care in the emergency hospitals, not least the hematology departments.

## Supporting information

**S1 Appendix. End-of-life questionnaire from the Swedish Register of Palliative Care.**
(PDF)

**S1 Table. Included variables.** All included variables from the end-of-life questionnaire, with frequencies of missing data and recoding information, when applicable.
(DOCX)

**S2 Table. All hematological diagnoses.** Place of death and access to specialized palliative care for all hematological diagnoses, with lymphoma subgroups.
(DOCX)

**S3 Table. Frequencies of co-morbidities.** Frequencies of co-morbidities in initial SRPC dataset of expected deaths.
(DOCX)

## Acknowledgments

The authors would like to acknowledge dr Staffan Lundström and the Swedish Register of Palliative Care for their support.

## Author contributions

**Conceptualization:** Per Fürst, Lena von Bahr.

**Formal analysis:** Ellen Skåreby, Lena von Bahr.

**Funding acquisition:** Lena von Bahr.

**Investigation:** Ellen Skåreby.

**Methodology:** Lena von Bahr.

**Supervision:** Lena von Bahr.

**Validation:** Per Fürst.

**Writing – original draft:** Ellen Skåreby.

**Writing – review & editing:** Per Fürst, Lena von Bahr.

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
