## [Decision Letter · Decision Letter 0]

22 Jan 2025

PONE-D-24-45924End-of-life care in hematological malignancies – a nationwide comparative study on the Swedish Register of Palliative CarePLOS ONE

Dear Dr. von Bahr,

Thank you for submitting your manuscript to PLOS ONE. After careful consideration, we feel that it has merit but does not fully meet PLOS ONE’s publication criteria as it currently stands. Therefore, we invite you to submit a revised version of the manuscript that addresses the points raised during the review process.

Dear Dr Bahr, please address the methodological and wording issues posed by Reviewer 2.

We look forward to receiving your revised manuscript.

Kind regards,

Rosemary Frey

Academic Editor

PLOS ONE

Journal Requirements:

This study received funding from Blodcancerfonden

4. Your abstract cannot contain citations. Please only include citations in the body text of the manuscript, and ensure that they remain in ascending numerical order on first mention.

Reviewers' comments:

Reviewer's Responses to Questions

**Comments to the Author**

1. Is the manuscript technically sound, and do the data support the conclusions?

Reviewer #1: Yes

Reviewer #2: Partly

2. Has the statistical analysis been performed appropriately and rigorously? 

Reviewer #1: Yes

Reviewer #2: I Don't Know

3. Have the authors made all data underlying the findings in their manuscript fully available?

Reviewer #1: No

Reviewer #2: Yes

4. Is the manuscript presented in an intelligible fashion and written in standard English?

Reviewer #1: Yes

Reviewer #2: Yes

5. Review Comments to the Author

Reviewer #1: Thank you for the opportunity to review this paper. I found this manuscript well-written and easy to read. The methodology used was appropriate, and the results clearly and logically presented. The grammar and spelling were of a high quality, however there are a few errors in the reference list. Also, the tables and figures in the supplementary material need labels. This research addresses an important issue, and highlights the lack of palliative care support faced by haematological cancer patients.

Reviewer #2: Thank you for this very important work, it is well needed to highlight the quality at end of life for this group.

Firstly in general regarding the conclusions and results; I think it would be better to "soften" the headings and the language describing the results/conclusions. Can we really say that an outcome is "impacted" by something? I would say "associated with" and/or correlated with depending on what analysis is done- we can not be sure that for example having a hematological diagnosis impacts anything, but is associated with. Throughout the whole manuscript you speak of impact- that is to me to draw the conclusions a bit far. I do understand what you mean, but a more strict language would be appropriate, since you show associations between for example death in an emergency hosptial and worse symptoms.

Another suggestion is to be more clear with you analysis and what has been done?

1. the Swedish palliative register - this is the last week in life? or how long back?

2. Please clarify in methods sections what co-morbidities was classified as exclusion criteria?

3. What were the dependent variables? Did you do a multivariate (several dependent variables) or a multiple regression analyis (one dependent variable)? It is very hard to follow how you did the analysis, which also leads to difficulties in interpreting the results. In line 104 there seems to be a word missing or a comma/period in the wrong place- not sure what is meant? In line 214 in results section- here it is very unclear how this was done? did you add this variable as a co-variate in the analysis-perhaps just how it is written.

I would like a reasoning on excluding unexpected deaths and co-morbidities? I think this is the real situation caring for patients with hematological malignancies, that sometimes death is quick and unexpected, both due to the disease itself, and also the treatment. Maybe clarify why not include also these patients? In line 266 in discussion, you draw the conclusion/reflection that since there was a documented end-of-life conversation, it could be a sign that they did not dy of curative treatment. But the line between curative and not is very fine in hematological patients, and it might be considered a curative treatment even if very slim chance- so not sure I follow this line of reasoning.

Another comment is that it could be that the unexpected deaths (that is excluded in you study), would be patients whith a very slim chance of cure, that died during or in complications of treatment, were totally "ignored" in terms of palliative needs, and this may be to a large extent impacting your results- so wish a more in depth method discussion on the choice to exclude these patients?(if I understood the meaning of an unexpected death in the right way)?

Furhter, line 295-7 in the discussion, is this really the only explanation? could it not be that for example pain assessment or oral assessments are done but not documented (thus not captured when registering in register?)- can we really assume that the missing data cohort have poorer quality of care at end of life? I agree with your reflections, but would be good to mention these institutions that do not even register in this registry?

I am overall in the discussion missing the reasoning on how to integrate palliative care in this group. We need to hava a palliative approach already during the treatment, and pursue both lines simultaneously- that is even a more challenge in my experience, and if we are going to provide high quality of end-of-life care for hematological patients in an emergency hospital, we need to talk about this. If a small chance of cure, what do we do and plan and hope for if this does not work? The solution can not be that as long as patients get palliative home care or hospice all is good.

In summary I welcome this important work, and with some clarifications this will be a good contribution to the body of research on patients with hematological malignancies and the need of palliative care/approach.

6. PLOS authors have the option to publish the peer review history of their article (what does this mean? ). If published, this will include your full peer review and any attached files.

**Do you want your identity to be public for this peer review?** For information about this choice, including consent withdrawal, please see our Privacy Policy .

Reviewer #1: No

Reviewer #2: No

---

## [Author Response · Author response to Decision Letter 1]

12 Feb 2025

Thank you for considering this paper for publication, and for the encouraging comments. We have tried to address all the issues raised, as detailed below. These answers are also submitted as a file "Response to reviewers"

First some responses to the editor, please let us know if anything is still unclear.

Role of funder: Blodcancerfonden as a funding agent had no role in study design, data collection and analysis, decision to publish, or preparation of the manuscript. Patient representatives mediated by Blodcancerfonden were involved in study design.

Regarding data sharing, we are unable to share the data due to the following reason as stated on submission:

The data contain potentially identifying information regarding individuals and therefore are subject to ethical and legal restriction to public sharing. We cannot share the data set as it is not permitted by Swedish law and the ethical permission obtained only allows for public sharing on a group level. Data are available from the Swedish Register of Palliative Care (info@palliativregistret.se) for researchers after appropriate ethical review.

Responses to reviewer 2:

Firstly in general regarding the conclusions and results; I think it would be better to "soften" the headings and the language describing the results/conclusions. Can we really say that an outcome is "impacted" by something? I would say "associated with" and/or correlated with depending on what analysis is done- we can not be sure that for example having a hematological diagnosis impacts anything, but is associated with. Throughout the whole manuscript you speak of impact- that is to me to draw the conclusions a bit far. I do understand what you mean, but a more strict language would be appropriate, since you show associations between for example death in an emergency hosptial and worse symptoms.

We agree, and have changed the wording in the manuscript to tone down the implications.

Another suggestion is to be more clear with you analysis and what has been done?

1. the Swedish palliative register - this is the last week in life? or how long back?

Yes, if not otherwise specified the variables are recorded for the last 7 days of life. This is now specified in the Methods section, row 112-113.

2. Please clarify in methods sections what co-morbidities was classified as exclusion criteria?

This is now included in Methods, row 103-105

3. What were the dependent variables? Did you do a multivariate (several dependent variables) or a multiple regression analyis (one dependent variable)? It is very hard to follow how you did the analysis, which also leads to difficulties in interpreting the results.

The method used was multiple regression analysis, using the glm() function in R with the outcome variable as dependent variable. This is now clarified in the Methods section, row 119.

In line 104 there seems to be a word missing or a comma/period in the wrong place- not sure what is meant?

This is a typing error and has been removed.

In line 214 in results section- here it is very unclear how this was done? did you add this variable as a co-variate in the analysis-perhaps just how it is written.

In this analysis we included iv fluids as a variable in the multiple regression analysis, in addition to age, sex and HM/ST. This has been clarified in the section.

I would like a reasoning on excluding unexpected deaths and co-morbidities? I think this is the real situation caring for patients with hematological malignancies, that sometimes death is quick and unexpected, both due to the disease itself, and also the treatment. Maybe clarify why not include also these patients?

In line 266 in discussion, you draw the conclusion/reflection that since there was a documented end-of-life conversation, it could be a sign that they did not dy of curative treatment. But the line between curative and not is very fine in hematological patients, and it might be considered a curative treatment even if very slim chance- so not sure I follow this line of reasoning.

Another comment is that it could be that the unexpected deaths (that is excluded in you study), would be patients whith a very slim chance of cure, that died during or in complications of treatment, were totally "ignored" in terms of palliative needs, and this may be to a large extent impacting your results- so wish a more in depth method discussion on the choice to exclude these patients?(if I understood the meaning of an unexpected death in the right way)?

Yes, this is very likely and we agree that the palliative needs of the hematological patients is probably under-estimated in this study. We have aimed for a conservative estimate, focusing on the patients where the conditions could be expected to be “optimal” for identifying and attending to palliative needs (expected deaths and no co-morbidities) as the nature of hematological diseases with “quick and unexpected” deaths are often used as an explanation for why hematological patients are less likely to be referred to palliative care. As we could show that even under optimal circumstances (as far as such can be identified in registry data) there are significant differences between hematological and solid tumor patients, we hope that this study can lead to discussions and possibly changes in practice in hematological care.

For unexpected deaths, the SPRC did not collect most of the variables of interest during the study period, and these cases were therefore not possible to include for most of the analyses.

Furhter, line 295-7 in the discussion, is this really the only explanation? could it not be that for example pain assessment or oral assessments are done but not documented (thus not captured when registering in register?)- can we really assume that the missing data cohort have poorer quality of care at end of life? I agree with your reflections, but would be good to mention these institutions that do not even register in this registry?

Yes, it is definitely possible, even likely, that assessments have been done but not documented. But that would happen in both HM and ST patients. This section in the discussion is only addressing the question whether the missing data is likely to be causing the significant differences reported, which would be the case if missing data could be expected to correlate with a higher likelihood of having made symptom assessments. We find this to be unlikely.

I am overall in the discussion missing the reasoning on how to integrate palliative care in this group. We need to hava a palliative approach already during the treatment, and pursue both lines simultaneously- that is even a more challenge in my experience, and if we are going to provide high quality of end-of-life care for hematological patients in an emergency hospital, we need to talk about this. If a small chance of cure, what do we do and plan and hope for if this does not work? The solution can not be that as long as patients get palliative home care or hospice all is good.

This is a very important point, and a discussion larger than this particular study. We have expanded the discussion in this regard, and hope for a continuation of that discussion in other forums.

In summary I welcome this important work, and with some clarifications this will be a good contribution to the body of research on patients with hematological malignancies and the need of palliative care/approach.

---

## [Decision Letter · Decision Letter 1]

24 Mar 2025

End-of-life care in hematological malignancies – a nationwide comparative study on the Swedish Register of Palliative Care

PONE-D-24-45924R1

Dear Dr., von Bahr,

We’re pleased to inform you that your manuscript has been judged scientifically suitable for publication and will be formally accepted for publication once it meets all outstanding technical requirements.

Kind regards,

Rosemary Frey

Academic Editor

PLOS ONE

Additional Editor Comments (optional):

Reviewers' comments:

Reviewer's Responses to Questions

**Comments to the Author**

1. If the authors have adequately addressed your comments raised in a previous round of review and you feel that this manuscript is now acceptable for publication, you may indicate that here to bypass the “Comments to the Author” section, enter your conflict of interest statement in the “Confidential to Editor” section, and submit your "Accept" recommendation.

Reviewer #1: All comments have been addressed

Reviewer #2: All comments have been addressed

2. Is the manuscript technically sound, and do the data support the conclusions?

Reviewer #1: Yes

Reviewer #2: Yes

3. Has the statistical analysis been performed appropriately and rigorously? 

Reviewer #1: Yes

Reviewer #2: Yes

4. Have the authors made all data underlying the findings in their manuscript fully available?

Reviewer #1: Yes

Reviewer #2: Yes

5. Is the manuscript presented in an intelligible fashion and written in standard English?

Reviewer #1: Yes

Reviewer #2: Yes

6. Review Comments to the Author

Reviewer #1: After reading through the revisied manuscript and reviewer responses, I am satisfied that the authors have addressed all reviewer comments. Thefore, I reccommend that the manuscript should be accepted for publication.

Reviewer #2: Thank you for you thorough replies and good luck in future. I look forward to more contribution to research on this group of patients.

7. PLOS authors have the option to publish the peer review history of their article (what does this mean? ). If published, this will include your full peer review and any attached files.

**Do you want your identity to be public for this peer review?** For information about this choice, including consent withdrawal, please see our Privacy Policy .

Reviewer #1: No

Reviewer #2: **Yes: ** Helena Ullgren

---

## [Editor Report · Acceptance letter]

PONE-D-24-45924R1

PLOS ONE

Dear Dr. von Bahr,

I'm pleased to inform you that your manuscript has been deemed suitable for publication in PLOS ONE. Congratulations! Your manuscript is now being handed over to our production team.

Kind regards,

on behalf of

Dr. Rosemary Frey

Academic Editor

PLOS ONE